# Chimeric Peptides Derived from Bovine Lactoferricin and Buforin II: Antifungal Activity against Reference Strains and Clinical Isolates of *Candida* spp.

**DOI:** 10.3390/antibiotics11111561

**Published:** 2022-11-05

**Authors:** Katherine Aguirre-Guataqui, Mateo Márquez-Torres, Héctor Manuel Pineda-Castañeda, Yerly Vargas-Casanova, Andrés Ceballos-Garzon, Zuly Jenny Rivera-Monroy, Javier Eduardo García-Castañeda, Claudia Marcela Parra-Giraldo

**Affiliations:** 1Human Proteomics and Mycosis Unit, Infectious Diseases Research Group, Department of Microbiology, Carrera 7 # 43-82 Lab 404 Building 52, Pontificia Universidad Javeriana, Bogotá 111321, Colombia; 2Bacteriology Department, Universidad Colegio Mayor de Cundinamarca, Bogotá 111321, Colombia; 3Synthesis and Application of Peptide Molecules Faculty of Sciences, Universidad Nacional of Colombia, Bogotá 111321, Colombia; 4MEDiS, INRAE, Université Clermont Auvergne, 63000 Clermont-Ferrand, France

**Keywords:** *Candida albicans*, bovine lactoferricin, Buforin II, antimicrobial peptides, chimeras

## Abstract

Antimicrobial peptides (AMPs) are considered to be a valuable source for the identification and/or design of promising candidates for the development of antifungal treatments, since they have advantages such as lower tendency to induce resistance, ease of production, and high purity and safety. Bovine lactoferricin (LfcinB) and Buforin II (BFII) are AMPs to which great antimicrobial potential has been attributed. The minimum motives with antimicrobial activity derived from LfcinB and BFII are RRWQWR and RLLR, respectively. Nine chimeras containing the minimum motives of both peptides were synthesized and their antifungal activity against fluconazole (FLC)-sensitive and resistant *C. albicans*, *C. glabrata*, and *C. auris* strains was evaluated. The results showed that peptides C9: (RRWQWR)_2_K-Ahx-RLLRRRLLR and C6: KKWQWK-Ahx-RLLRRLLR exhibited the greatest antifungal activity against two strains of *C. albicans*, a FLC-sensitive reference strain and a FLC-resistant clinical isolate; no medically significant results were observed with the other chimeras evaluated (MIC ~200 μg/mL). The chimera C6 was also active against sensitive and resistant strains of *C. glabrata* and *C. auris*. The combination of branched polyvalent chimeras together with FLC showed a synergistic effect against *C. albicans.* In addition to exhibiting antifungal activity against reference strains and clinical isolates of *Candida* spp., they also showed antibacterial activity against both Gram-positive and Gram-negative bacteria, suggesting that these chimeras exhibit a broad antimicrobial spectrum and can be considered to be promising molecules for therapeutic applications.

## 1. Introduction

Invasive candidiasis (IC) is the most prevalent fungal disease in intensive care unit (ICU) patients, accounting for 75% of all fungal infections. IC affects about 250,000 people each year, and its mortality rate is estimated to be 70% [1,2]. In the United States, candidemia is the third leading cause of bloodstream infections in ICU patients [3].

In these scenarios, the most frequently isolated yeast is *Candida albicans*, causing about 60% of all genital, oral, and cutaneous candidiasis infections [4]. *C. albicans* has been described as an innocuous commensal microorganism that is part of the microbiota of the gastrointestinal tract, genitourinary tract, oral cavities, and conjunctiva. When the host is immunosuppressed and/or has microbiota imbalance, this yeast can cause superficial infections (skin, mucous membranes of the mouth, and vagina). The hematogenous spread of the fungus can lead to invasive infections in almost all organs, which without effective treatment are life-threatening [5].

On the other hand, in recent years, other non-albicans species have been gaining importance; for example, in the United States and Europe, the second most isolated yeast is *C. glabrata*, whose incidence varies between 7% and 15% and commonly affects neoplastic and elderly patients [1,6,7]. Furthermore, the emerging yeast *C. auris* is clinically relevant, because it presents mortality rates ranging from 30% to 60% in immunocompromised patients and can additionally exhibit multidrug-resistance profiles [8,9].

The increase in fungal infections caused by *Candida* is mainly due to an increase in patients with neoplastic disease and/or who are HIV positive, and to prolonged use of medical devices (tubes, catheters, prostheses, and valves) and resistance caused by overuse of antibiotics [10]. Conventional antifungal agents have some disadvantages, such as toxicity and low efficacy against resistant strains, among others. For instance, amphotericin B (AmB) causes acute renal toxicity after prolonged administration, and FLC and itraconazole have low efficacy against some yeasts and cause hepatotoxicity [11,12].

AMPs are considered to be an important source of promising molecules for the development of new antifungal treatments. AMPs are part of the innate immune response, are less likely to induce adverse effects, are safe, and have a broad spectrum of activity [13,14]. On the other hand, AMPs have pharmacokinetic limitations, such as low bioavailability, due primarily to factors such as susceptibility to proteases, difficulty crossing membranes, rapid elimination from the body, etc. [15]. To overcome these limits and improve the antimicrobial activity, the native sequence of AMPs has been modified. Some of the optimization strategies of AMPs include reducing the amino acid chain, inserting non-natural amino acids, increasing the polyvalence of the sequence, and combining sequences of two or more AMPs (chimeras), among others [16,17,18,19,20]. LfcinB is a peptide fragment of 25 residues in length located in the N-terminal region of bovine lactoferrin (LfB) and is generated during hydrolysis of this protein with gastric pepsin [21,22]. It has been found that LfcinB possesses antimicrobial activity against fungi, bacteria, and viruses, among others [23,24,25]. The mechanism of action described for LfcinB is related to its electrostatic interaction between the cationic side chains and the negatively charged components of the cell surface of microorganisms, so the side chain of hydrophobic residues, such as tryptophan, interact with the lipid bilayer, destabilizing the cell membrane and causing lysis [21]. Buforin II (BFII) is a 21-residue peptide that is generated by the treatment of BFI with Lys-C endoproteinase. BFII causes permeability of the cell membrane, producing transient toroidal pores without generating lysis of the microorganism. After its internalization, it interacts with nucleic acids (DNA and RNA), inducing cell death [26,27].

LfcinB (20–25): RRWQWR is the shortest sequence derived from LfcinB with antibacterial, antifungal, and anticancer activity. In similar way, Buforin II (BFII) is an antimicrobial peptide with activity against Gram-positive and Gram-negative bacteria as well as fungi. Palindromic sequences have been designed from the 4-residue motif (RLLR) of Buforin II, and their antibacterial activity has been evaluated against Gram-negative and Gram-positive strains. The palindrome [RLLR]_5_: RLLRRLLRRLLRRLLRRLLR was reported to have an MIC of 0.7 µM against *E. coli.* Chimeras containing the minimum motif RRWQWR attached to RLLR or RLLRRRLLR showed greater antibacterial activity than individual sequences [18,20,27,28,29,30].

In the present study, the antifungal activity of chimeric peptides containing the minimal motifs of both LfcinB (RRWQWR) [28,29] and BFII (RLLR) [27,30] against reference strains and clinical isolates of *C. albicans, C. glabrata*, and *C. auris,* sensitive and resistant to FLC, was evaluated. Three chimeric peptides with antifungal activity against these strains were identified. The synergistic effect on the antifungal activity of mixing some chimera with FLC was also established.

## 2. Results and Discussion

The chimeras used in this work contain sequences derived from LfcinB, an AMP that affects the integrity of the cell membrane, and from BFII, an AMP that is internalized into the cell and causes DNA damage. The antifungal activity of the chimeras was initially evaluated against the reference strain *C. albicans* ATCC SC5314, sensitive to FLC and the clinical isolate of *C. albicans* 256 HUSI-PUJ, which is resistant to FLC. Considering their primary structure and polyvalence, the chimeras evaluated in this study were classified into three groups (Table 1):(I)The C1 and C3 peptides were synthesized and evaluated in order to establish whether the antifungal activity is affected by the position of the RRWQWR or RLLR sequences in the chimera. Chimeras C2 and C4 contained Ahx (6-aminohexanoic acid residue) as a spacer for the two motifs. This is intended to establish whether the inclusion of the Ahx spacer between the two motifs affects the antifungal activity. The inclusion of the Ahx spacer facilitates the synthesis of the chimera and separates the two motifs so that each of them can interact independently with the cell surface of the pathogen.(II)Chimeras C5, C6, and C7 were synthesized and their antifungal activity was evaluated in order to determine whether partial or total replacement of Arg residues with Lys in RRWQWR and/or RLLR sequences affects their antifungal activity. Replacing Arg residues with Lys has been shown to facilitate and reduce the cost of chimeric synthesis [18].(III)Chimeras C8 and C9 were synthesized in order to establish whether the polyvalence of the RRWQWR motif increased the antifungal activity. Previous reports have shown that the polyvalence of the RRWQWR sequence increases antibacterial and anticancer activity [20,31].

**Table 1 antibiotics-11-01561-t001:** MIC and MFC values for each branched chimeric in the two *C. albicans* strains.

Antifungal Activity Against *C. albicans* Strains. µg/mL (µM)						
Group	Code	Sequence	ATCC SC5314		256 HUSI-PUJ	
			MIC	MFC	MIC	MFC
Control	LfcinB (20–25)	RRWQWR	200 (203)	200 (203)	200 (203)	200 (203)
	BFII (32–35)_Pal_	RLLRRLLR	>200 (>183)	>200 (>183)	>200 (>183)	>200 (>183)
I	C1	RRWQWRRLLR	200 (131)	>200 (>131)	100 (66)	>200 (>131)
	C2	RRWQWR-Ahx-RLLR	200 (122)	>200 (>122)	200 (122)	>200(>122)
	C3	RLLRRRWQWR	100 (66)	200 (131)	100 (66)	200 (131)
	C4	RLLR-Ahx-RRWQWR	>200 (>122)	>200 (>122)	200 (122)	>200 (>122)
II	C5	RRWQWR-Ahx-KLLKKLLK	100 (48)	200 (97)	100 (48)	200 (97)
	C6	KKWQWK-Ahx-RLLRRLLR	50 (24)	100 (48)	50 (24)	100 (48)
	C7	KKWQWK-Ahx-KLLKKLLK	200 (101)	>200 (>101)	200 (101)	>200 (>101)
III	C8	(RRWQWR)_2_K-Ahx-RLLR	100 (37)	100 (37)	100 (37)	100 (37)
	C9	(RRWQWR)_2_K-Ahx-RLLRRLLR	50 (15)	50 (15)	50 (15)	50 (15)

### 2.1. Minimum Inhibitory and Fungicidal Concentration

In this investigation, chimeric peptides containing the precursor sequences LfcinB (20–25): RRWQWR and BFII (32–35)_Pal_: RLLRRLLR were synthesized and purified, and their antifungal activity against reference strain *C. albicans* ATCC SC5314 sensitive to FLC (MIC= 1 μg/mL) and a clinical isolate of *C. albicans* 256 HUSI-PUJ resistant to FLC (MIC = 64 μg/mL) was evaluated. Antifungal activity of the peptides RLLR or RLLRRRLLR has not been reported in the literature consulted to date.

The precursor peptides (controls) exhibited the lowest antifungal activity against *C. albicans* SC5314 and *C. albicans* 256 (Table 1). Peptide LfcinB (20–25) showed MIC and MFC values of 200 µg/mL (203 µM), while peptide BFII (32–35)_Pal_ had MIC and MFC values > 200 µg/mL, (>183 µM) greater than the maximum concentration of the peptide evaluated, indicating that these peptides do not exert significant antifungal activity against these strains. This is consistent with the results obtained by Muñoz et al. [32] and Pineda et al. [18] when they evaluated the peptide RRWQWR against the yeast *Saccharomyces cerevisiae* FY1679 and the peptide RLLRRLLR against the gram-positive bacteria *Staphylococcus aureus* ATCC 25923 and *Enterococcus faecalis* ATCC 29212, where they did not find any antimicrobial activity at the concentrations evaluated (48–200 µg/mL).

In group I, the C1 (RRWQWRRLLR) and C3 (RLLRRWQWR) chimeras lacked the Ahx spacer and the RRWQWR and RLLR motifs were alternatively bound in the N-terminal or C-terminal region of the sequence (Table 1). Moreover, the C1 and C3 chimeras showed higher antifungal activity against *C. albicans* strain SC5314 and the FLC-resistant clinical isolate *C. albicans* 256 than the precursor peptides and the C2 and C4 chimeras, suggesting that the inclusion of Ahx affects the antifungal activity against this strain. It was previously reported that the C1 and C3 chimeras showed antibacterial activity against Gram-positive and Gram-negative ATCC bacterial strains [18], suggesting that these chimeric peptides exhibit a broad spectrum of antimicrobial action. The C3 chimera showed higher antifungal activity (MIC ~100 µg/mL) against *C. albicans* SC5314 and the clinical isolate *C. albicans* 256 than the C1 peptide, suggesting that the position of the LfcinB (20–25) and BFII (32–35) motifs in the sequence might be relevant for antifungal activity against *C. albicans.* These results suggest that chemical linkage of the two precursor sequences in the C3 chimera enhances antifungal activity against the two *C. albicans* strains tested.

In group II, peptides C5 (RRWQWR-Ahx-KLLKKLLK) and C6 (KKWQWK-Ahx-RLLRRLLR) exhinited greater antifungal activity against the two strains evaluated than peptides C1, C2, C3, C4, and C7 (Table 1). These results indicate that the substitution of Arg with Lys in the RRWQWR sequence significantly increased the antifungal activity, and the substitution of the Arg residues with Lys residues in the RLLRRLLR motif also increased the antifungal activity against both strains. On the other hand, the substitution of all Arg residues with Lys in chimera C7 did not increase the antifungal activity (MIC 200 µg/mL, 101 µM). Moreover, the results reported for C5 and C6 are similar to those obtained by Pineda et al.: C5 (MIC 100 µg/Ml, 48 µM) and C6 (MIC 50 µg/mL,24 µM) showed greater antibacterial activity than chimera C2 (MIC 200 µg/mL,122 µM) [18,33]. According to the above, although for chimera C5 and C7 the MIC values can be considered similar to those of the precursors (controls), the C6 chimera exhibited greater activity, obtaining a MIC value up to 4 times lower than RRWQWR and RLLRRLLR peptides.

In group III, peptides C8 ((RRWQWR)_2_K-Ahx-RLLR) and C9 ((RRWQWR)_2_K-Ahx-RLLRRLLR) exhibited antifungal activity against the strains evaluated. Peptide C9 exhibited the greatest antifungal activity against both strains of *C. albicans*. These results suggest that the polyvalence of the motif RRWQWR plus the lineal dimer of the sequence RLLR increased the antifungal activity (Table 1).

Peptides C3, C5, C6, C8, and C9 showed moderate antifungal activity against the reference strain *C. albicans* SC5314 and the clinically isolated *C. albicans* 256, resistant to FLC. According to the categories proposed by Alves et al., synthetic antifungal molecules with MIC values between 26–100 μg/mL are considered to have moderate antifungal activity [34].

The results suggest that the partial replacement of Arg with Lys in one of the two motifs or the polyvalence of the RRWQWR motif enhances the antifungal activity. Peptides C6 and C9 showed the best results; both peptides exhibited the highest antifungal activity against the strains evaluated. Peptide C6 has more advantages for synthesis than peptide C9, due to the fact that peptide C6 has lesser residues than peptide C9; however, peptide C9 had the lowest MIC value against the clinically isolated *C. albicans* 256 resistant to FLC.

The results above described are in agreement with previous reports, in which peptides containing the minimal motif of LfcinB presented MICs between 0.8 and 400 µg/mL for *C. albicans* [35]. In addition, Chang et al. [36] reported that the viability of cells treated with LfcinB decreased as the concentration of LfcinB increased; similarly, the antifungal activity of the chimera was concentration dependent. Furthermore, Jang et al. [37] reported that peptides containing the minimum motif of BFII showed antifungal activity, with MIC values of 32 to >64 μg/mL in strains of *C. albicans* and *Cryptococcus neoformans*. This is the first report describing the antifungal activity of chimeras containing sequences of two AMPs (LfcinB and Buforin II). It is possible that chimeras composed of two AMP sequences exert their antifungal activity through the combination of their mechanisms of action. It has been suggested that LfcinB acts by permeating the cell membrane [21]. On the other hand, BFII is internalized and interacts with nucleic acids by inhibiting processes of replication of genetic material and protein translation, leading to cell death [38]. The branched chimeric peptide C9 (MIC = 50 µg/mL/15 μM) exhibited greater antifungal effect compared to linear chimeras C2 (MIC = 200 µg/mL/122 μM) and C5 (MIC = 100 µg/mL/48 μM), possessing the same linker as the branched chimeras.

Taking into account the definition of fungistatic activity (decrease in the growth of a fungus by <99%) and fungicidal activity (decrease in the growth of a fungus by ≥99%) [39], our results show that chimeras C5, C6, C8, and C9 exhibited fungistatic and fungicidal activity in *C. albicans* SC5314 and *C. albicans* 256 (Figure 1). When a molecule exhibits fungicidal and/or fungistatic activity, therapeutic and prophylactic options, which can be monitored clinically, increase. It is important to highlight that chimeras C5, C6, C8, and C9 inhibited the growth of the *C. albicans* strains evaluated by more than 90%, being that these strains belong to the species most frequently involved in invasive candidiasis forms. The effect generated by the chimeric peptides depends on the concentration used; thus, it can be seen that higher concentrations are required to achieve the fungicidal effect compared to the concentrations that cause fungistatic activity (Table 2).

Chimeras C5 and C6 showed fungicidal and fungistatic effects on *C. albicans* SC5314 and *C. albicans* 256 strains. Chimera C5 at 100 µg/mL (48 µM) showed a fungicidal effect against the *C. albicans* SC5314 strain during 48 h of incubation and a fungistatic effect at a chimera concentration of 50 µg/mL (24 µM) (Figure 1a). When the FLC-resistant clinically isolated *C. albicans* 256 was incubated for 48 h with chimera C5 at 200 µg/mL (97 µM), a fungicidal effect was observed, while chimera C5 at 100 µg/mL (48 µM) induced a fungistatic effect, inhibiting yeast growth up to approximately 30 h of incubation (Figure 1b).

Chimera C6 at 50 µg/mL (24 µM) exhibited a fungicidal effect against *C. albicans* SC5314, while at 25 ug/mL (12 µM), the effect was fungistatic, a decrease in the exponential phase being observed (Figure 1c). Similarly, this chimera at 100 µg/mL (48 µM) completely inhibited the growth of *C. albicans* 256 for 48 h. When this strain was incubated with chimera C6 at 50 µg/mL (24 µM), a fungistatic effect was observed (Figure 1d).

Chimera C8 generated a fungicidal effect on *C. albicans* SC5314 at 200 µg/mL (73 µM) during 48 h of treatment, while at 100 µg/mL (37 µM), a fungistatic effect was observed, and yeast growth was completely inhibited up to about 23 h (Figure 1e). Similarly, chimera C8 at 200 µg/mL (73 µM) also showed a fungicidal effect against *C. albicans* 256 after 48 h of treatment, while at 100 µg/mL (37 µM), it completely inhibited the yeast growth up to nearly 40 h of incubation (Figure 1f).

Chimera C9 exhibited fungicidal activity at 25 µg/mL (8 µM) on both *C. albicans* strains during 48 h of incubation, while at 12.5 µg/mL (4 µM) it exerted a fungistatic effect on the strains evaluated; the strain growth was completely inhibited up to approximately 20 h of treatment (Figure 1g,h). These results suggest that the antifungal activity is enhanced by the joining of two polyvalent motifs, the linear dimer of the RLLR sequence and the branched dimer of the RRWQWR.

However, *C. albicans* 256 (resistant to FLC) and *C. albicans* SC5314 (sensitive to FLC) exhibited susceptibility to chimeras C3, C5, C6, C8, and C9, suggesting that the antifungal activity of both peptide chimeras is not affected by the resistance mechanisms that the isolated strains employ against FLC, a conventional antifungal. This particularity is of great relevance, since these chimeras could be considered promising candidates for developing potential treatments against even resistant yeasts.

Additionally, the antifungal activity of C6 was assessed against other clinically relevant species, i.e., *C. glabrata* and *C. auris* [40,41,42] (Figure 2). Two sensitive (*C. glabrata* ATCC2001 and *C. auris* 0001) and two resistant (*C. glabrata* 1875 caspofungin-resistant and *C. auris* 537 AmB- FLC-resistant) isolates were included. Regarding the *C. glabrata* results, interestingly, in the reference strain, at 24 µM (50 µg/mL), C6 exerted a fungicidal effect, whereas the caspofungin-resistant isolate, at a concentration of 192 µM (400 µg/mL), exerted a fungistatic effect. For *C. auris*, the C6 chimera at 192 µM (400 µg/mL) showed a fungistatic effect on both isolates.

### 2.2. Hemolytic Effect

Chimeras C3, C5, and C6 did not show a significant hemolytic effect (2–4%) at MIC concentrations, indicating that these peptides are not toxic to normal erythrocytes. The precursor peptide RLLR (200 μg/mL) exhibited a hemolytic effect of 63%, and the peptides RRWQWR (200 µg/mL) and RLLRRLLR (200 µg/mL) exerted a hemolytic effect of 1% and 2%, respectively, while chimeras containing the motif RLLR or RLLRRLLR did not show a significant hemolytic effect at MIC concentrations (hemolytic activity data of chimeras C1 to C7 were previously published by [18]). Similarly, the C8 chimera did not show a hemolytic effect on erythrocytes at any peptide concentration evaluated, even at concentrations where a fungicidal effect occurs (200 μg/mL), indicating that this peptide is not toxic to human red blood cells at the concentrations evaluated. Chimera C9, at 12.5 μg/mL (4 μM) and 25 μg/mL (8 μM), did not show hemolysis; at these chimeric concentrations, fungistatic and fungicidal effects, respectively, were observed (Table 3). Our results suggest that the inclusion of sequences with hemolytic activity in chimeras improves antifungal activity and decreases the hemolytic effect.

### 2.3. Antifungal Activity of Chimeric Peptides Mixed with FLC

The mixture of chimera C6 (25 μg/mL) and FLC (0.5 μg/mL) showed an additive effect on antifungal activity (FICI = 1) in *C. albicans* SC5314, showing a twofold reduction in the MIC values of the peptide and the FLC. In the FLC-resistant clinical isolate *C. albicans* 256, the combination of peptide C6 and FLC induced an indifferent effect (FICI = 1.03). These results suggest that the combination of the C6 chimera with FLC, although it did not exhibit a synergistic effect against *C. albicans* SC5314, can increase the activity of FLC by a factor of two when mixed with a C6 chimera, while no significant antifungal activity was evidenced for the FLC-resistant strain *C. albicans* 256 (Table 4).

In contrast, a synergistic relationship was observed when both branched chimeric peptides (C8 and C9) were evaluated in combination with FLC in *C. albicans* SC5314, decreasing the MIC by a factor of between four and eight for the peptides and four for FLC (Table 4); this feature makes it possible to potentiate the antifungal activity of both molecules. The FLC-sensitive and -resistant *C. albicans* SC5314 and 256 strains, respectively, show equal susceptibility to each of the branched chimeras, evidence that the antifungal activity of the evaluated peptides is not affected by conventional antifungal resistance mechanisms.

The C8 peptide combined with FLC showed synergy for the antifungal activity against *C. albicans* SC5314, decreasing the MIC of the peptide by a factor of eight and a factor of four for FLC; however, the combination of the C8 peptide with FLC induces an indifferent effect. *C. albicans* 256 (Table 4). These results suggest that the synthesis of chimeras containing MAP sequences is a promising strategy for combating resistant fungal mycoses, in order to extend the shelf life and efficacy of currently used drugs through the use of combination therapy. The results suggest that FLC-combined chimeras have the advantage of producing antifungal activity against resistant strains and may also alter the fungistatic activity of many compounds [43].

The emergence of the multidrug-resistant yeast *C. auris* and the circulation of resistant clones *C. glabrata* make it imperative to search for new therapeutic alternatives [44,45,46]. The C6 chimera was evaluated in combination with FLC against *C. glabrata* ATCC 2001 and 1875 and *C. auris* 001 and 537, obtaining an additive effect in three of them. For the reference strain *C. glabrata* 2001, an additive effect (FICI: 0.6) was observed, decreasing the MIC of the peptide by half and the FLC up to 10 times, in a way similar to what occurred in clinical isolates *C. auris* 001 and C. auris 537. In both cases, the MIC of FLC decreased by a factor of 2. Finally, for *C. glabrata*, an indifferent effect was obtained (Table 5).

The results obtained by combining AMPs with FLC are consistent with the effect found by Vargas et al. [47], who also observed an additive effect against *C. albicans* and *C. auris* by the combination of a palindromic peptide derived from LfcinB with FLC, achieving an 8-fold enhancement of the activity of the AMPs through this mixture.

## 3. Materials and Methods

### 3.1. Reagents and Materials

The strain *C. albicans* SC5314 was purchased from ATCC (Manassas, VA, USA), and the clinical isolate *C. albicans* 256 HUSI-PUJ was obtained from the oral mucosa of a patient in the San Ignacio Hospital and deposited in the strain bank of the MICOH-P group of the PUJ (Pontificia Universidad Javeriana). Sabouraud dextrose agar (SDA), Roswell Park Memorial Institute 1640 Medium (RPMI 1640), saline, sterile distilled water (H_2_Od), seeding loop, O+ human red blood cells, 5 mL tubes with EDTA anticoagulant, Tween 20, Saline 0. 85% saline, NEST Petri dishes, 96-well flat bottom and U-bottom test plates, fluconazole, red blood cells 50 mL Falcon tubes, 10 mL Falcon tubes, multichannel pipettes, 20–200 µL pipettes, 2–20 µL pipettes, 200 µL yellow tips, 200 µL yellow tips, 2–20 µL yellow tips, 200 µL yellow tips, 100–1000 µL blue tips, 0.1–10 µL white tips, 2 mL Eppendorf tubes, laminar flow chamber, ELISA reader (Expert Plus ASYS), spectophotometer, Bioscreen C. 100-well honey comb plates specific for Bioscreen C. Fmoc-Arg(Pbf)–OH, Fmoc-Trp(Boc)–OH, Fmoc-Gln(Trt)–OH, Fmoc-Leu-OH, Fmoc-Lys(Fmoc)–OH, Fmoc-6-Ahx-OH, Rink amide resin, dicyclohexilcarbodiimide (DCC), and 1-hydroxy-6-Chlorobenzotriazole were purchased from AAPPTec (Louisville, KY, USA). Trifluoroacetic acid (TFA), acetonitrile (ACN), dichloromethane (DCM), N,N-dimethylformamide, ethanodithiol, triisopropylsilane, methanol, acetonitrile, and isopropanol were obtained from Merck (Darmstadt, Germany). SPE SupelcleanTM columns were purchased from Sigma-Aldrich (St. Louis, MO, USA).

### 3.2. Peptides

Chimeric peptides containing the LfcinB and BFII motifs were (i) synthesized using solid-phase peptide synthesis using Fmoc/tBu strategy, (ii) purified using RP-SPE chromatography, and (iii) characterized by RP-HPLC and MS (Supplementary material), following the protocol reported by [19].

### 3.3. In Vitro Antifungal Susceptibility Test

Antifungal susceptibility testing was carried out using the broth microdilution (BMD) method, following the CLSI M27-A3 guidelines with slight modifications [48]. Briefly, cells (0.5–2.5 × 10^3^ CFU/mL) were incubated with peptides (200, 100, 50, 25, 12.5, and 6.25 µg/mL) at 37 °C for 48 h. MICs were visualized and densitometry (595 nm, microplate reader, Bio-Rad, ᵢMark^TM^) was used to determine the lowest concentration of peptide that caused a significant decrease (MIC/2 or ≥50%) compared with the growth control (cells incubated in absence of peptide) (n = 3). In order to verify that the peptides were able to kill the yeast cells, the plates were also evaluated for minimum fungicidal concentration (MFC). Briefly, aliquots from each well from susceptibility testing assays were transferred to plates containing Sabouraud dextrose agar (SDA), which were then incubated at 37 °C for 24 h. The highest dilution with no growth on the agar plate was considered to be the MFC.

### 3.4. Time-Kill Curves

A time-kill kinetic assay was carried out according to the method previously described by Pfaller and coworkers [49] with minor modifications. The peptides (C5, C6, C8, and C9) diluted in RPMI were tested at a range of concentrations: 0 (control), 0.2, 0.5, 1, and 2 times the MIC value for each strain. The isolates were subcultured on SBD, and then an inoculum was adjusted to a 0.5–2.5 × 10^3^ CFU/mL in an RPMI 1640 medium. Yeast inoculum (150 uL) was added to a 100-well plate containing serial dilutions of the peptides. The plates were incubated with agitation at 37 °C in a plate reader (Bioscreen C MBR automated turbidometric analyzer, Growth Curves Ltd., Helsinki, Finland), which takes hourly absorbance readings. Moreover, fluconazole MICs were used as a control. The readings were analyzed with Bioscreen software (Growth Curves USA, Piscataway, NJ, USA) and GraphPad Prism 8.0.1 (GraphPad Soft-ware, San Diego, CA, USA). Statistical comparisons were made using an analysis of variance (two-way ANOVA) followed by a Tukey–Kramer post hoc test. P values less than 0.005 were considered significant. To determine the fungistatic and fungicidal effects, a growth inhibition of >70% for 48 h was considered fungistatic, and a yeast kill of ≥99% for 72 h was considered fungicidal. Moreover, previously determined FLC MICs (data not shown) were used as a control.

### 3.5. Hemolysis Assays

The hemolysis assays were carried out following the methodology reported by Vargas et al. [20] with some modifications, as follows. First, 5 mL of heparinized peripheral blood was centrifuged at 1000 g for 7 min. The erythrocyte fraction was suspended in 10 mL of saline solution (SS) and washed twice by centrifugation at 1000 g for 7 min. The erythrocytes (4% hematocrit) were incubated with peptide (ranging from 6.2 to 200 μg/mL), for 2 h at 37 °C. SS was used as negative control, while distilled water was used as a positive control. The mixtures were centrifuged, the supernatants were collected, and the absorbance was determined to be 450 nm.

### 3.6. Checkerboard Test

The in vitro interactions between the peptides (C6, C8, and C9) and the FLC drug were evaluated in a checkerboard assay, as previously described by Cokol et al. [50]. Briefly, each isolate was prepared by picking colonies from an overnight culture in SBD at 37 °C and suspended in a sterile normal saline PBS buffer. The fungal inoculum was adjusted to 0.5 MacFarland and the inoculum was adjusted to 0.5–2.5 × 10^3^ CFU/mL. The drug-peptide combinations were formed over a range of concentrations: 0 (control) and 0.06 to 2 times the MIC [51]. The fractional inhibitory concentration index (FICI) was calculated as follows:



$$\mathrm{FICI}=\frac{\mathrm{CML}\;\mathrm{peptide}\;\mathrm{in}\;\mathrm{combination}}{\mathrm{CMI}\;\mathrm{peptide}\;\mathrm{alone}}+\frac{\mathrm{CML}\;\mathrm{FLC}\;\mathrm{in}\;\mathrm{combination}}{\mathrm{CMI}\;\mathrm{FLC}\;\mathrm{alone}}$$


The calculated FICI was reported as synergy with values FICI ≤ 0.5, additive FICI > 0.5–1, indifference FICI > 1–4, and antagonism FICI ≥ 4 [52].

A synergistic effect was considered to be when the effect of the combination exceeded the effects of the individual components. An additive effect was when the effect of the combination was equal to the sum of the effects of the individual components. An indifferent effect was one in which the activity was equal to the effects of the most active component. Finally, an antagonism was a reduced effect of a drug combination observed compared to the effect of the single most effective substance [52].

## 4. Conclusions

In this study, we report for the first time the antifungal activity of nine chimeras containing the LfcinB and BFII minimal motifs. The chimeras C9: (RRWQWR)_2_K-Ahx-RLLRRLLR and C6: KKWQWK-Ahx-RLLRRLLR exhibited the highest antifungal activity (MIC 50 µg/mL) than the precursors, no medically significant results were observed (MIC ~200 µg/mL) for the other chimeras evaluated. The peptide C9 showed promising antifungal properties being both fungistatic and fungicidal (25–12.5 µg/mL) against *C. albicans*. We also observed that the inclusion of Ahx as a spacer affects the antifungal activity against this strain. Furthermore, changing Arg to Lys in the LfcinB or BFII motifs enhanced the antifungal activity of the chimeras against the reference strain *C. albicans* SC5314 and the clinically isolated FLC-resistant *C. albicans* 256. The chimera containing the dimeric peptide LfcinB showed the highest antifungal activity against the tested strains, suggesting that this chimera design could be a successful strategy to enhance antifungal activity against reference strains or clinical isolates of *Candida* spp. It was also observed that these modifications help to reduce the toxicity of the peptides against mammalian cells, such as erythrocytes. Finally, it could be demonstrated that the chimeric peptides could be considered as a therapeutic alternative that acts individually, but also when combined with antifungals such as fluconazole.

## Figures and Tables

**Figure 1 antibiotics-11-01561-f001:**
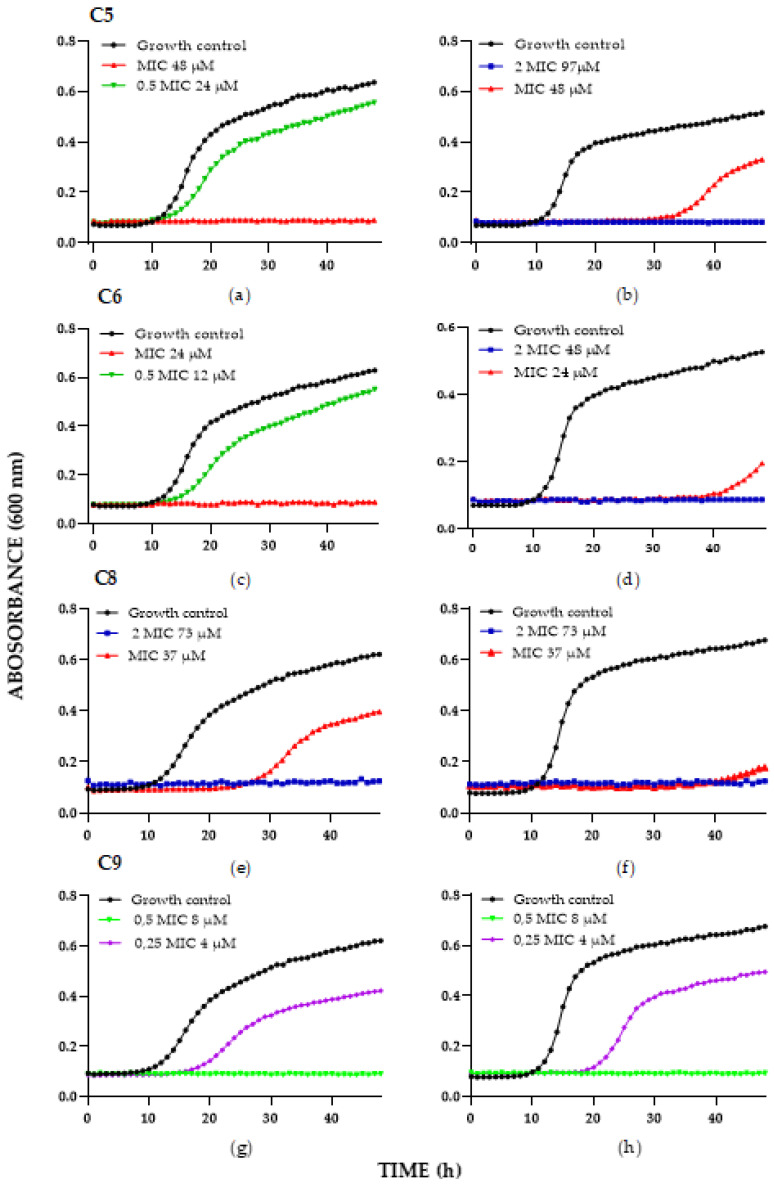
Time-kill curve of chimeras against reference and clinical isolates *Candida* species: (**a**,**c**,**e**,**g**) *C. albicans* SC5314, (**b**,**d**,**f**,**h**) C. *albicans* 256. The MIC value corresponds to the concentration obtained by the broth concentration method.

**Figure 2 antibiotics-11-01561-f002:**
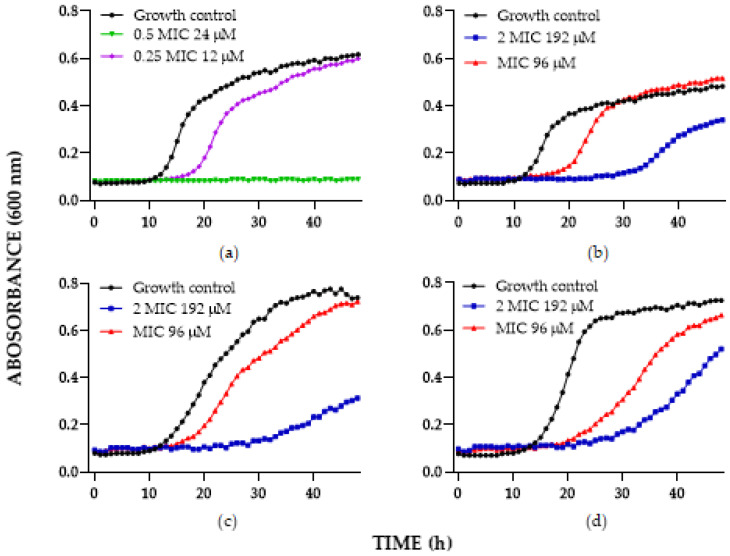
Time-kill curve of chimera C6 against: (**a**) *C. glabrata* 2001, (**b**) *C. glabrata* 1875, (**c**) *C. auris* 001, (**d**) *C. auris* 537. The MIC value corresponds to the concentration obtained by the broth concentration method.

**Table 2 antibiotics-11-01561-t002:** Fungistatic and fungicide effect values for chimeras with higher activity against *C. albicans*.

Time-Kill Result against *C. albicans* Strains—µg/mL (µM)				
Peptide	ATCC SC5314		256 HUSI-PUJ	
	Fungistatic *	Fungicide *	Fungistatic *	Fungicide *
C5	50 (24)	100 (48)	100 (48)	200 (97)
C6	25 (12)	50 (24)	50 (24)	100 (48)
C8	100 (37)	200 (73)	100 (37)	200 (73)
C9	13 (4)	25 (8)	13 (4)	25 (8)

* The values correspond to the fungistatic and fungicidal activity after 48 h of incubation.

**Table 3 antibiotics-11-01561-t003:** Hemolysis percentage for control peptides and peptide chimeras.

Hemolytic Activity		
Peptide	Concentration (μg/mL) *	% Hemolysis
RLLR	200	63
BFII (32–35)_Pal_	200	2
LfcinB (20–25)	200	1
C1	100–200	5
C2	200	3
C3	100–200	4
C4	200	5
C5	50–200	2
C6	25–100	2
C7	200	6
C8	100	2
C9	13–50	2–11

* The values correspond to the concentrations where antifungal activity was evidenced.

**Table 4 antibiotics-11-01561-t004:** Effect of combining the chimeric peptides with FLC on *C. albicans* SC5314 and 256 HUSI/PUJ strains.

Synergistic Result *C. albicans*								
*C. albicans* Strain	Peptide	MIC_a_	MIC_b_	A	B	FICI	MIC_a_/A	MIC_b_/B
ATCC SC5314	C6	50	1	25	0.5	1	2	2
	C8	200	0.5	25	0.13	0.38	8	4
	C9	100	0.5	25	0.13	0.5	4	4
256 HUSI-PUJ	C6	100	32	3.1	32	1.03	32	1
	C8	100	32	50	32	1.5	2	1
	C9	50	32	25	16	1	2	2

MICa and MICb correspond to the MIC (µg/mL) of the chimeric peptide and fluconazole, respectively, and A and B are the MIC values when combining the peptides and fluconazole. Minimum fractional concentration index (FICI), MICa/A, and MICb/B represent the factor by which the chimera or FLC are potentiated after being evaluated in combination, respectively.

**Table 5 antibiotics-11-01561-t005:** Effect of combining the C6 chimeric with FLC on *C. glabrata* and *C. auris* strain.

Synergistic Result *C. glabrata* and *C. Auris*							
Strain	MIC_a_	MIC_b_	A	B	FICI	MIC_a_/A	MIC_b_/B
*C. glabrata* ATCC 2001	100	0.3	50	0.03	0.6	2	10
*C. glabrata* 1875 CHU-PUJ	400	4	12.5	4	1.03	32	1
*C. auris* 001 HUSI-PUJ	400	32	12.5	16	0.53	32	2
*C. auris* 537 HUSI-PUJ	400	64	200	32	1	2	2

## Data Availability

Not applicable.

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
