# Peer review of "Chimeric Peptides Derived from Bovine Lactoferricin and Buforin II: Antifungal Activity against Reference Strains and Clinical Isolates of Candida spp."

_antibiotics, 2022, doi:10.3390/antibiotics11111561_

Round 1
Reviewer 1 Report
In this manuscript the authors described the synthesis and chemical and biological characterisation of chimeric peptides. These chimeric peptides containing short peptides derived from LfcinB and Buforin II. Some chimeric peptides showed antifungal activity against a fluconazole sensitive and resistant strain of C. albicans. Using different arrangement of the two peptide motifs in the chimeric peptides some structure-activity relationships were determined.
Remarks and questions
1. It is not clear why these short sequences were selected for the synthesis of chimeric peptides. It should be explained in the introduction part with references.
2. The design of the peptides is not clear. What is the reason that the structure of chimeric peptides with spacer was modified further? Based on the Table 1 peptides without spacer are better. Why BFII (32-35)Pal and its derivative were used in peptide Group II, or was not it used in Group I? In Group II peptide C6 was the best construct with KKWQWK sequence, but this was not used in Group III. The peptide C5, C6, C8 and C9 should be modified based on the structure-activity relationship, and peptide RRWQWRKLLKKLLK, KKWQWKRLLRRLLR, (KKWQWK)2KRLLR, (KKWQWK)2KRLLRRLLR should be examined.
Reviewer 2 Report
The negative results are also very important and accordingly should be published. The authors are too optimistic about their results and I would encourage them to address the issue of very high MIC values appropriately in the text. The reader gets impression that the peptides are of great medical importance while in fact only one is somewhat interesting. The more interesting issues I find is the use of AMPs with FLC. The authors should provide an additional control, i.e. use two peptides LfcinB and BFII administered together (check the toxicity as well). Please follow my comments below. I would consider the article for publications but only after the issues below are addressed.
Line 21 - 26
It should be clearly written that the antifungal activity of most investigated peptides are of no medical significance (MIC ~200 ug/ml). Only the peptide C9 showed some promising activity against fungi. I find the abstract somewhat misleading. It should be changed to better reflect the content of the performed research.
Line 89 – 91
This information about C. albicans is already provided in the introduction. Delete it form the section results.
Line 92 – 95
This should be in the section Materials and Methods.
Line 95 -97
The sentence does not make sense: ’The antifungal activity of these chimeras was initially evaluated against the reference strain C. albicans ATCC SC5314 and is sensitive to FLC and the clinical isolate of C. albicans 256 HUSI-PUJ, which is resistant to FLC.’ The reference strain C. albicans ATCC SC5314 is sensitive to FLC not the antifungal activity!
Line 111
Please provide citation.
Line 114 – 116
Please provide citation.
Line 123 – 124
As an additional control, I would also use two peptides LfcinB and BFII administered together.
Line 133 148
I would abbreviate the paragraph and made it more concrete. Please note that the MIC values are very high even for the C3 peptide.
Line 151 – 154
First, the complete substitution of Arg with Lys made the peptide inactive against the investigated fungi (MIC 200ug/ml), which was not discussed at all. Only partial substitution of Arg to Lys showed increased antifungal activity! Third, there should be a control peptide ‘RRWQWR-Ahx-RLLRRLLR’ investigated. The whole paragraph requires clarification and please indicate that the MIC values are still very high for all II group peptides.
Line 158 – 159
Please indicate that the MIC values are still very high.
Line 166 – 157
I would add ‘…showed some antifungal activity…’ – please note that the MIC values are still very high. These peptides are of no medical significance.
Line 167 – 169
This information should be in Line 151 – 154.
Line 183 – 190
I would strongly advice to make an additional experiment and investigate the use of the two peptides LfcinB and BFII administered together and compare it with the peptide C9. We should also compare C9 not with C5 but ‘RRWQWR-Ahx-RLLRRLLR’.
Line 207 – 247
Be consistent which concentration you give in parentheses throughout the article, sometimes ug/ml sometimes uM.
Line 312- 322
Numbers should be on the right of the page.
Line 411 – 421
Many paragraphs in the section Results were too detailed in the description. However, the conclusions are too short and simplistic. Please indicate that the MIC was generally very high and that only one peptide C9 actually showed promising antifungal properties. Your created three sets of peptides, conclude the results here, i.e. the role of partial replacement of Arg to Lys, Ahx spacer and polyvalence. Then go to toxicity and the additive effect.
Reviewer 3 Report
In this work the authors design and synthesize nine chimeras containing the minimum motives of both LfcinB and BFII. They also investigate the antifungal activity of them against several Candida strains. And find that peptide C9 and C6 exhibited the greatest antifungal activity against two strains. This study will fetch interest among researchers in antimicrobial peptides, especially for AFP.
Major and minor comments to the authors are as follows, which may further improve the quality of this manuscript:
Line 76-77, components of the cell surface of microorganisms.
Please provides the RP-HPLC and MS of each synthesized peptide as supplementary materials, also information such as net charge, pI, calculated MW and measured MW, etc., of each peptide.
How to determine and confirm the minimal motifs of LfcinB and BFII? Please explain it.
Line 117-122, please indicates MFC of FLC against both tested strains.
Line 123-132, BFII seems to be not active against both C. albicans strains, even at the highest tested concentration. The authors conclude that this is consistent with the results obtained by references. But the reference is concern of the activity against bacteria, not fungi.
Line 142, ATCC changes as bacteria?
Line 143-148, C3 (BFII-LfcinB) showed greater antifungal activity than peptide C1. Why not design peptides based on this template next?
Line 173-176, the sentence seems out of context here.
For Fig 1 and Fig 2, how to determine the used MIC for time-kill assay?
Please describes the 2.4 results briefly.
In Table 5, 12,5 changes as 12.5, etc.
Round 2
Reviewer 1 Report
I thank the authors for the answers. Unfortunately I still do not see the rational design of the chimeras.
In my opinion is that the spacer may facilitate the synthesis, but it is not necessary for it. So, it does not explain why constructs without spacer were not studied.
Based on the activity of group I, it can be said that the spacer decreases the activity of chimeras and not only affect their activity (lines 150 and 425). In this group C3 was the best, but this arrangement of peptides was not used in case of group II. Why?
Because the control peptide, RRWQWR-Ahx-RLLRRLLR, is missing in group II we cannot say that the Arg-Lys replacement increases the activity (line 161-162), and “polyvalence of the motif RRWQWR plus the lineal dimer of the sequence RLLR increased the antifungal activity” (line 175-177). Why was not this chimera studied? Its activity is necessary for the correct interpretation of the results.
Based on the activity of group I and II, peptides (KKWQWK)2KRLLRRLLR and RLLRRLLRK(KKWQWK) KKWQWK should be more active, but they were not examined.
Author Response
Reviewer 1
Comment: I thank the authors for the answers. Unfortunately, I still do not see the rational design of the chimeras. In my opinion is that the spacer may facilitate the synthesis, but it is not necessary for it. So, it does not explain why constructs without spacer were not studied. Based on the activity of group I, it can be said that the spacer decreases the activity of chimeras and not only affect their activity (lines 150 and 425). In this group C3 was the best, but this arrangement of peptides was not used in case of group II. Why? Because the control peptide, RRWQWR-Ahx-RLLRRLLR, is missing in group II we cannot say that the Arg-Lys replacement increases the activity (line 161-162), and “polyvalence of the motif RRWQWR plus the lineal dimer of the sequence RLLR increased the antifungal activity” (line 175-177). Why was not this chimera studied? Its activity is necessary for the correct interpretation of the results. Based on the activity of group I and II, peptides (KKWQWK)2KRLLRRLLR and RLLRRLLRK(KKWQWK) KKWQWK should be more active, but they were not examined.
Answer. We thanks to the reviewer for the comments. We would like to clarify that:
- For this research, all control and chimeric peptides were obtained at the same time, and the goal of each group of chimeric peptides was described in the manuscript (lanes 111-117).
- After obtaining the molecules, we performed the antifungal activity of the chimeras and the control peptides, and it should be pointed out that the antifungal activity of Group I was not used for design Group II or III.
- As the reviewer mentioned, it is interesting to evaluated the antifungal activity of molecules as the dimer (KKWQWK)2KRLLRRLLR. We consider this molecule for future experiments, actually we have tried to synthesised this dimer but it is highly polar and it was difficult to purify the product because its hygroscopic; so this problem did not allow us to evaluated its antifungal activity. In this context, new lysine-rich dimer peptides are a synthetic challenge and we are working in solve this problem.
We consider that our manuscript shows to the reader the use of lineal and dimeric chimeric peptides, for getting new antifungal candidates. this is a starting point and all the valuable reviewer suggestions will be considered for designing the next step of our research

Reviewer 2 Report
English should be corrected in the modified paragraphs of the manuscript, eg. Line 104 and 106 “which is” is not necessary, line 192 “, are” is not necessary etc.
Author Response
Reviewer 2
Comment: English should be corrected in the modified paragraphs of the manuscript, eg. Line 104 and 106 “which is” is not necessary, line 192 “, are” is not necessary etc.
Answer: We thanks to the reviewer for his/her comments, we check the paragraphs and corrected them
Reviewer 3 Report
According to this version, no more comments were provided.
Author Response
We thanks to the reviewer for his/her comments